# Optimal Lower Bounds and New Upper Bounds for Sequential Prediction with Abstention

## Abstract

We study the problem of sequential prediction with abstentions, where a learner faces a stream of i.i.d. data interspersed with adversarial examples, a setting introduced by [GHMS24]. The learner can abstain, incurring no penalty on adversarial points, but is penalized for mistakes and for abstaining on i.i.d. points. Prior work left open whether a fundamental gap exists between the known and unknown distribution settings, and their positive results for the unknown case were restricted to simple classes whose structure they heavily exploited. We resolve both of these questions. First, we establish an $\Omega(\sqrt{T})$ lower bound on the error for any learner facing a VC-dimension 1 class when the distribution is unknown, proving the existing algorithms are optimal and demonstrating a quantitative separation from the logarithmic error achievable when the distribution is known. Second, we provide the first sublinear error algorithm for a more complex geometric class, achieving an $\tilde{O}(T^{2/3})$ error bound for biased half-spaces in $\mathbb{R}^2$.

## 1 Introduction

Sequential prediction models often face a trade-off between robustness to adversarial examples and performance on stochastic data. While standard online learning algorithms perform well in i.i.d. settings, their guarantees degrade in the presence of an adversary. To address this, [GHMS24] recently introduced a framework for sequential prediction that allows a learner to abstain. In their model, an adversary can inject out-of-distribution examples into a stream of i.i.d. data. By abstaining, the learner can avoid making high-risk predictions on these adversarial inputs. The learner's error is measured as the sum of the number of misclassifications and the number of abstentions on i.i.d. examples; abstentions on adversarial examples incur no penalty.

The work of [GHMS24] established foundational results in this setting for a time horizon $T$. They demonstrated that if the underlying i.i.d. distribution is known, an algorithm can achieve an error bound of $O(d^2 \log T)$ for concept classes with VC dimension $d$. When the distribution is unknown, they provided algorithms achieving $O(\sqrt{dT})$ error for VC-dimension 1 classes ($d = 1$) and for axis-aligned rectangles in $\mathbb{R}^d$ with a corner at the origin. Both of these algorithms heavily exploit the simple combinatorial structure of their respective classes, a property not shared by more general classes like half-spaces.

However, their work leaves two critical questions unanswered: (1) *Is the gap between the $O(\log T)$ error in the known-distribution setting and the $\sqrt{T}$ error in the unknown-distribution setting fundamental?*, and (2) *Can we design efficient algorithms with sublinear loss for more complex and natural concept classes in the unknown distribution setting?*

This paper resolves both of these questions. Our main contribution are:

Submitted to 39th Conference on Neural Information Processing Systems (NeurIPS 2025). Do not distribute.

1. We establish the first separation between the known and unknown distribution settings by proving an $\Omega(\sqrt{T})$ lower bound on the error rate for a VC-dimension 1 concept class. This proves the optimality of the existing algorithm for this class and confirms that knowledge of the underlying distribution provides a fundamental advantage.

2. We present the first algorithm for learning the natural concept class of biased half-spaces in $\mathbb{R}^2$ in the unknown distribution setting. Specifically, our algorithm achieves a mistake bound of $\tilde{O}(T^{2/3})$, demonstrating that sublinear loss is possible for more complex geometric classes.

**Other Related Work**

**PQ Learning** Another line of work dealing with learning with a mixture of iid and adversarial samples is PQ learning [GSSV24][GKKM20]. This setting differs from the one studied here primarily because they assume the learner is given an entirely non-adversarial training set.

**Clean Labeled Data Poisoning** [BHQS21] studies the offline learning setting, where an adversary adds cleanly labeled adversarial points to the training set. The idea of attackability is used in the paper, and is also a crucial component of the abstention error upper bounds in this paper and those in [GHMS24]. They also show that half-spaces can not be learned in their setting. The crucial difference that allows us to learn half-spaces in our setting is due to the definition of the rate of learning. In the lower bounds from [BHQS21] the rate of learning is with respect to the number of non-adversarial samples, while we instead use the number of total samples (including adversarial samples). Intuitively, we allow the adversary to trick the learner on a sample, so long as the learner gets many more samples they predict correctly on.

For more discussion of other related models, see [GHMS24].

# 2  Preliminaries

We start by formalizing the learning model and introducing relevant concepts from learning theory.

**Notation.** Let $\mathcal{X}$ be the domain or instance space. A concept class $\mathcal{F}$ is a set of functions $f : \mathcal{X} \to \{-1, 1\}$. We work in the realizable setting, where labels are generated by an unknown target function $f^* \in \mathcal{F}$. Let $\mathcal{D}$ be a distribution over $\mathcal{X}$.

**The Learning Model.** We consider the sequential prediction model with abstention from [GHMS24]. The interaction between the learner and the adversary proceeds in rounds for a time horizon of $T$. At the beginning, an adversary chooses a distribution $\mathcal{D}$ over $\mathcal{X}$ and a target function $f^* \in \mathcal{F}$. In each round $t = 1, \ldots, T$:

1. The adversary decides whether to inject an adversarial example. It chooses $q_t \in \{0, 1\}$. If $q_t = 0$, nature draws $x_t \sim \mathcal{D}$. If $q_t = 1$, the adversary chooses an arbitrary $x_t \in \mathcal{X}$ to send to the learner.

2. The learner receives $x_t$ and outputs a prediction $\hat{y}_t \in \{-1, 1, \bot\}$, $\bot$ denotes abstention.

3. The learner receives the true label $y_t = f^*(x_t)$.

The learner's goal is to minimize its total error, which is the sum of two quantities: the misclassification error and the abstention error.

$$Err_{mis} := \sum_{t=1}^{T} \mathbf{1}[\hat{y}_t \neq y_t \wedge \hat{y}_t \neq \bot], \qquad Err_{abs} := \sum_{t=1}^{T} \mathbf{1}[\hat{y}_t = \bot \wedge q_t = 0]$$

The total error is the sum of these two terms. Note that the learner is not penalized for abstaining on adversarial examples ($q_t = 1$).

**VC Dimension.** The complexity of the concept classes we consider will be measured by the Vapnik-Chervonenkis (VC) dimension.

**Definition 1** (Shattering and VC Dimension)**.** *A concept class $\mathcal{F}$ is said to* shatter *a set of points $S = \{x_1, \ldots, x_k\} \subseteq \mathcal{X}$ if for every possible labeling $b \in \{-1, 1\}^k$, there exists a function $f \in \mathcal{F}$ such that $(f(x_1), \ldots, f(x_k)) = b$. The* VC dimension *of $\mathcal{F}$, denoted* VCDim$(\mathcal{F})$, *is the size of the largest set shattered by $\mathcal{F}$.*

## 3 Lowerbound

We first define the concept class and adversary strategy used to establish the lower bound.

**Concept Class.** Fix $B := \lfloor \sqrt{T} \rfloor$. The domain $\mathcal{X}$ is the set of nodes in a complete $B$-ary tree of depth $B$. For each leaf $\theta \in [B]^B$, corresponding to a unique root-to-leaf path, we define a function $f_\theta : X \to \{-1, 1\}$ as:

$$f_\theta(v) = \begin{cases} 1 & \text{if } v \text{ is a prefix of the path } \theta, \\ -1 & \text{otherwise.} \end{cases}$$

The concept class is the set of all such functions, $F = \{f_\theta : \theta \in [B]^B\}$. This class has $\mathsf{VCdim}(F) = 1$, as any single point can be labeled in two ways, but no two points can be shattered[1].

**Adversary.** The adversary's strategy is defined over $B$ blocks, each of $B$ rounds. First, the adversary samples a secret path $\theta \sim \mathrm{Unif}([B]^B)$ and a non-adversarial block index $r \sim \mathrm{Unif}([B])$.

- In each block $i \in [B]$, the adversary defines a distribution $D_i$ as the uniform distribution over the $B$ children of the depth-$(i-1)$ prefix of $\theta$. Under the true concept $f_\theta$, exactly one of these points is labeled 1 (can think of it as a "singleton"), and the other $B-1$ points are labeled $-1$.

- For all rounds $t$ within block $i$, the adversary provides a sample $x_t \sim D_i$. For the remaining $T - B^2$ rounds, the adversary can play arbitrarily.

- The round is non-adversarial ($c_t = 0$) if $i = r$ implying $\mathcal{D} = D_r$, and an adversarial injection ($c_t = 1$) otherwise.

Crucially, the sequence of examples drawn by the adversary is statistically independent of the choice of the non-adversarial block $r$, therefore the learner has no way to learn $r$.

**Theorem 1.** *For the adversary above, any learner's expected total error is lower bounded by:*

$$\mathbb{E}[Err_{mis}] + \mathbb{E}[Err_{abs}] \geq \left( \frac{(1 - e^{-1})^2}{2} \right) B - O(1) = \Omega(\sqrt{T}).$$

**Proof Sketch.** Within each block, up until the learner discovers the singleton, for each unique sample the learner sees, predicting $-$ is wrong with probability around $1/B$, predicting $+$ is wrong with probability around $1 - 1/B$, and abstaining will incur abstention error with probability $1/B$. Since each block is $B$ and sampling uniformly over $B$ elements, the number of unique samples within the block is $\Omega(B)$, so the minimum expected error per block incurred is at least $\Omega(1)$. Summing over all of the blocks gives the desired bound of $\Omega(B)$.

For the full proof, see appendix A.

## 4 Upperbound for Half-spaces in $\mathbb{R}^2$

We will consider learning biased half-spaces in 2 dimensions, VC-dimension 3 classs. Formally, $\mathcal{X} = \mathbb{R}^2$, and $\mathcal{F} = \{f \mid \exists w, b \in \mathbb{R}^2 \text{ st } f(x) = (w^\top x + b > 0) \lor f(x) = (w^\top x + b \geq 0)\}$. [2]

**Learner.** Our learner is a generalization of the learner for VC-dimension 1 classes in [GHMS24]. At a high level, all of the learners for this setting (including the one for axis-aligned rectangles in [GHMS24]) guess if being incorrect tells the learner something about the past samples they have recieved. For our learner (and the VC-dimension 1 learner), sets of past examples can 'vote' for the learner to predict $x$ has some label $\ell$ if being wrong permanently decreases the number of ways that set can be labeled conditioned on the labels of the past examples with the label $\ell$. It is not too hard to show that this algorithm gives a mistake bound for any setting with a finite number of labels, but it is more difficult to show that it can simultaneously bound the number of incorrect abstentions. This is done in [GHMS24] by exploiting structure inherent to all VC-dimension 1 classes, and we will do it by exploiting the geometric structure of half spaces.

For each label $\ell \in \{1, -1\}$, define $\Phi_\ell : \mathcal{X}^* \times \mathcal{X}^* \to \mathbb{N}$ as $\Phi_\ell(U, S) = \left| \{f_U \mid f \in \mathcal{F}_{|S \to \ell}\} \right|$.

---

[1]Interestingly, any VC dimension 1 class is equivalent to prefixes of some rooted tree [BD15].

[2]Half-spaces in 1 dimension essentially reduce to thresholds which have been explored in prior work (see [GHMS24] for discussion).

Where $f_U$ is $f$ with its domain restricted to $U$, and $\mathcal{F}_{|S \to \ell} = \{f' \in \mathcal{F} \mid f'(x) = \ell \ \forall x \in S\}$. In other words, for any set of samples $U$, and set of labeled samples $S$, $\Phi_\ell(U, S)$ counts the number of distinct ways to label $U$ while being consistent with labeling all of $S$ as $\ell$.

Define the voting function used by the learner

$$\rho_\ell(x, S, U) = \sum_{(a,b) \in S} \left(\Phi_\ell(\{a,b\}, U) - \Phi_\ell(\{a,b\}, U \cup \{x\})\right)$$

The learner is defined in algorithm 1.

---

**Algorithm 1:** Learner for half-spaces in $\mathbb{R}^2$

---

Set $S^+ = \emptyset, S^- = \emptyset, S = \emptyset$
**for** $t = 1, \dots, T$ **do**
    Receive $x_t$
    **if** $S^+ = \emptyset$ **then** predict $\hat{y}_t = -1$
    **else**
        **if** $S^- = \emptyset$ **then** predict $\hat{y}_t = 1$
    **else**
        **if** $\exists \ell \in \{-1, 1\}$ *st $\hat{y}_t = \ell$ is inconsistent with $S$* **then** predict $\hat{y}_t$ consistent with $S$
        **else**
            **if** $\max_{\ell \in \{-1,1\}} \rho_{-\ell}(x, S^\ell, S^{-\ell}) > \alpha$ **then** predict
            $\hat{y}_t = \texttt{argmax}_{\ell \in \{-1,1\}} \rho_{-\ell}(x, S^\ell, S^{-\ell})$
            **else** predict $\hat{y}_t = \perp$
    Upon receiving label $y_t$, update $S^{y_t} \leftarrow S^{y_t} \cup \{x_t\}$, and $S \leftarrow S \cup \{(x_t, y_t)\}$.

---

**Theorem 2.** *If $\mathcal{F}$ is biased half-spaces in $\mathbb{R}^2$, then algorithm 1 achieves*

$$\mathbb{E}[Err_{mis}] = O\left(\frac{T^2}{\alpha}\right) \qquad \mathbb{E}[Err_{abs}] = O(\sqrt{\alpha} \ln T)$$

Note that with $\alpha = T^{4/3}$, we can bound both types of error by $\tilde{O}(T^{2/3})$.

**Proof Sketch.** The main idea is in two parts, one for the mistake bound, and one of the abstention bound, both using $\alpha$ but in inverse ways. each voting pair can only vote incorrectly a small number of times, so we can bound the number of mistakes using the voting threshold ($\alpha$) and the total number of voting pairs. For abstention error, we show that for every set of samples of size four, either one of them would not be abstained upon by the learner, or one of them is voted for by a pair of the others, regardless of the what the adversary does. This allows us to show that good probability (controlled by $\alpha$) the next iid samples will receive sufficient votes to not be abstained upon.

A higher $\alpha$ reduces the number of mistakes the learner can make, but also increases the likelihood of abstaining on iid samples, by choosing the correct value to balance them, we can make both sublinear.

For the full proof, see appendix B.

# 5 Conclusion

We showed that there exist simple concept classes which can not be learned with error under $\Omega(\sqrt{T})$ when the iid distribution is unknown, as opposed to being able to learn with error $O(\ln T)$ for finite VC-dimension. We also showed that half-spaces in $\mathbb{R}^2$ are learnable with error $O(T^{2/3})$.

A promising direction for future work is to extend the algorithm for half-spaces to work in $\mathbb{R}^n$, the most simple modification could possibly achieve an error of $T^{1-1/(n+1)}$. It remains open whether half-spaces can be solved with error $O(\sqrt{T})$.

Characterizing learnability in the unknown distribution setting remains an open problem. Although some finite VC-dimension classes are not learnable with logarithmic error, it is unknown whether all are learnable with sublinear error.

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

# A  Proof of theorem 1

*Proof.* Since the within-block streams are i.i.d. from $D_i$ regardless of the non-adversarial block index $r$, $r$ is independent of the transcript and

$$\mathbb{E}[\text{Abstention Error}] = \mathbb{E}[A_r] = \frac{1}{B} \sum_{i=1}^{B} \mathbb{E}[A_i]. \qquad (1)$$

We fix an arbitrary (possibly randomized) learner and then fix its internal randomness (a standard application of Yao's minimax principle), making its behavior deterministic for the analysis.

**Per-block analysis using the "not-seen" state.** Fix a block $i$. Let $m \in \{1, \dots, B\}$ be the number of distinct support points that appear among its $B$ draws. We order these distinct points by their first encounter, $k = 1, \dots, m$. Let $\tau \in \{1, \dots, m, m+1\}$ be the first-encounter index at which the singleton appears, let $\tau = m+1$ if the singleton does not appear. Conditional on the unlabeled sequence of first encounters and the singleton appearing, $\tau$ is uniform on $[m]$. The singleton appears with probability $m/B$, so with probability $1 - m/B$, $\tau = m+1$.

For each $k \in [m]$, we define the learner's *pre-singleton action* $a_k^\circ \in \{+1, -1, \bot\}$ to be the action the learner takes at the $k$-th new point given the history in which the first $k-1$ first-encounters all had label $-1$. This action is well-defined and depends only on the unlabeled sequence and the fixed internal randomness, not on the true value of $\tau$. On the event $\{\tau \geq k\}$, the actual action at the $k$-th first encounter equals $a_k^\circ$. We only count errors forced by the singleton's appearance:

- If $a_k^\circ = +1$: a mistake is made whenever $\tau > k$, contributing an expected error of $\Pr(\tau > k \mid m) = \frac{m}{B}(m-k)/m + (1 - \frac{m}{B}) = (B-k)/B$.

- If $a_k^\circ = -1$: a mistake is made exactly when $\tau = k$, contributing an expected error of $\Pr(\tau = k \mid m) = \frac{m}{B}1/m = 1/B$.

- If $a_k^\circ = \bot$: an abstention occurs whenever $\tau \geq k$. Via (1), this contributes $\Pr(\tau \geq k \mid m) \cdot (1/B) = \frac{\frac{m}{B}(m-k+1)/m+1-m/B}{B} = \frac{B-k+1}{B^2}$ to the total expected error.

The total expected error, conditioned on $m$ and the pre-singleton actions $(a_k^\circ)$, is the sum of these costs over all steps. We can lower-bound the total error by summing these minimum costs at each step. Recall that $A_i$ is the abstention error incurred at round $i$, and $M_i$ is the misclassification error.

The total expected error for the block is the expectation over $\tau$ of the errors forced by its position:

$$\mathbb{E}\left[\sum_{k=1}^{B} M_i + \frac{A_i}{B} \mid m, (a_k^\circ)\right] \geq \mathbb{E}_\tau\left[\sum_{k=1}^{\tau-1} \mathbf{1}\{a_k^\circ = +1\} + \mathbf{1}\{a_\tau^\circ = -1\} + \frac{1}{B}\sum_{k=1}^{\tau} \mathbf{1}\{a_k^\circ = \perp\}\right]$$

$$= \sum_{k=1}^{m} \left(\Pr(\tau > k)\mathbf{1}\{a_k^\circ = +1\} + \Pr(\tau = k)\mathbf{1}\{a_k^\circ = -1\} + \frac{\Pr(\tau \geq k)}{B}\mathbf{1}\{a_k^\circ = \perp\}\right)$$

$$= \sum_{k=1}^{m} \left(\frac{B-k}{B}\mathbf{1}\{a_k^\circ = +1\} + \frac{1}{B}\mathbf{1}\{a_k^\circ = -1\} + \frac{B-k+1}{B^2}\mathbf{1}\{a_k^\circ = \perp\}\right).$$

For any strategy, the cost is at least the sum of the point-wise minimums:

$$\mathbb{E}\left[M_i + \frac{A_i}{B} \mid m\right] \geq \sum_{k=1}^{m} \min\left\{\frac{B-k}{B}, \frac{1}{B}, \frac{B-k+1}{B^2}\right\}. \tag{2}$$

Let $j := B - k + 1$ (re-indexing from the last encounter). The term inside the sum becomes $\frac{1}{B}\min\{j-1, 1, j/B\}$. For $j = 1$ (i.e., $k = B$), the minimum is 0. For $j \geq 2$ and $j \leq B$, we have $j/B \leq 1$ and $j/B \leq j - 1$, so the minimum is $j/B$. The bound becomes:

$$\mathbb{E}\left[M_i + \frac{A_i}{B} \mid m\right] \geq \sum_{j=2}^{m} \frac{j}{B^2} = \frac{m(m+1)/2 - 1}{B^2}.$$

**From one block to all blocks.** Since $\mathbb{E}[m(m+1)/2 - 1] \geq \mathbb{E}[m]^2/2 - 1$ for all $m \geq 0$, taking the expectation over $m$ yields:

$$\mathbb{E}[M_i] + \frac{1}{B}\mathbb{E}[A_i] \geq \mathbb{E}\left[\frac{m(m+1)/2 - 1}{B^2}\right] \geq \frac{\mathbb{E}[m]^2/2 - 1}{B^2} = \frac{\mathbb{E}[m]^2}{2B^2} - \frac{1}{B^2}.$$

The expected number of distinct points is $\mathbb{E}[m] = B(1 - (1 - 1/B)^B) \geq B(1 - e^{-1})$. Thus,

$$\mathbb{E}[M_i] + \frac{1}{B}\mathbb{E}[A_i] \geq \frac{B^2(1 - e^{-1})^2}{2B^2} - \frac{1}{B^2} = \frac{(1 - e^{-1})^2}{2} - \frac{1}{2B^2}.$$

Summing over all $B$ blocks and using (1), we have $\sum_i \mathbb{E}[M_i] = \mathbb{E}[Err_{mis}]$ and $\sum_i \mathbb{E}[A_i]/B = \mathbb{E}[Err_{abs}]$. Therefore,

$$\mathbb{E}[Err_{mis}] + \mathbb{E}[Err_{abs}] \geq \sum_{i=1}^{B}\left(\frac{(1 - e^{-1})^2}{2} - \frac{1}{B^2}\right) = B\left(\frac{(1 - e^{-1})^2}{2}\right) - 1.$$

This gives the desired $\Omega(\sqrt{T})$ bound. $\qquad\qquad\square$

# B Proof of theorem 2

*Proof.* Notice that while $S^+ = \emptyset$ or $S^- = \emptyset$, the learner can make at most 2 mistakes, and will never abstain, so we will consider what happens after that initial phase.

**Misclassification Error.** Let $t_1, \ldots t_m$ denote all the times when the learner makes a misclassification error. Each time $t_i$ must have $\rho_{-y_{t_i}}(x, S_{t_i}^{-y_{t_i}}, S_{t_i}^{y_{t_i}}) \geq \alpha$, since the learner did not abstain, so

$$\sum_{i=1}^{m} \rho_{-y_{t_i}}(x, S_{t_i}^{-y_{t_i}}, S_{t_i}^{y_{t_i}}) \geq \sum_{i=1}^{m} \alpha = \alpha Err_{mis}$$

Recall that $\Phi_{-\ell}(\{a, b\}, S_i^{-\ell})$, which counts the number of ways $a, b$ can be labeled conditioned on $S_i^{-\ell}$ being labeled $-\ell$. notice that $\Phi_{-\ell}(\{a, b\}, S_i^{-\ell}) \in \{1, 2, 3, 4\}$ for any valid $S^{-\ell}$, and also it can

never increase when $S^{-\ell}$ grows. We can use this to make a complementary bound:

$$\sum_{i=1}^{m} \rho_{-y_{t_i}}(x, S_{t_i}^{-y_{t_i}}, S_{t_i}^{y_{t_i}}) \leq \sum_{t=1}^{T} \rho_{-y_t}(x, S_t^{-y_t}, S_t^{y_t})$$

$$= \sum_{t=1}^{T} \sum_{(a,b)\in S_t^{-y_t}} (\Phi_{y_t}(\{a,b\}, S_t^{y_t}) - \Phi_{y_t}(\{a,b\}, S_t^{y_t} \cup \{x\}))$$

$$= \sum_{t=1}^{T} \sum_{(a,b)\in S_t^{-y_t}} (\Phi_{y_t}(\{a,b\}, S_t^{y_t}) - \Phi_{y_t}(\{a,b\}, S_{t+1}^{y_t}))$$

$$= \sum_{\ell\in\{-1,+1\}} \sum_{(a,b)\in S_t^{\ell}} \sum_{t\in T | y_t \neq \ell} (\Phi_{-\ell}(\{a,b\}, S_t^{-\ell}) - \Phi_{-\ell}(\{a,b\}, S_{t+1}^{-\ell}))$$

$$= \sum_{\ell\in\{-1,+1\}} \sum_{(a,b)\in S_t^{\ell}} (\Phi_{-\ell}(\{a,b\}, S_0^{-\ell}) - \Phi_{-\ell}(\{a,b\}, S_T^{-\ell}))$$

$$\leq \sum_{\ell\in\{-1,+1\}} \sum_{(a,b)\in S_t^{\ell}} (4-1)$$

$$= 3\left(\binom{|S^+|}{2} + \binom{|S^-|}{2}\right) < \frac{3}{2}T^2$$

The second to last step follows since there are at most $4$ ways to label the pair $a, b$, and always at least one (when conditioning on realizable labels).[3]

Combining these two bounds gives us that the number of misclassifications is at most $O(\frac{T^2}{\alpha})$.

**Abstention Error.** We will first use an attackability argument to bound the probability of the learner abstaining on the $i$th iid sample of label $\ell$.

**Definition 2.** *For any hypothesis $f$, and history $S \subseteq \mathcal{X}$ a sample $x \in S$ is attackable if there exists a set $S' \subseteq \mathcal{X}$ such that the learner abstains on $x$ when given the history $S' \cup S \setminus x$ labeled by $f$.*

We will show that any set of samples $S$ has at most $2\alpha$ attackable samples.

Let $S$ be partitioned into $S^+$ and $S^-$ based on the labels.

For either $\ell \in \{+1, -1\}$, consider any $\{x_0, x_1, x_2, x_3\} \in \left(S_{iid}^{\ell}\right)^4$.

By Radons theorem there exist a partition of $\{x_0, x_1, x_2, x_3\}$ into disjoint subsets $U, T$ such that $conv(U) \cap conv(T) \neq \emptyset$. There are two cases, in the first, one of the sets, has only one element, then that element is clearly not attackable. In the other case, both $U, T$ have two elements. Say without loss of generality that $U = \{x_0, x_1\}$ and $w^\top x_0 \leq w^\top x_i$. Then consider any $x^4 \in S^{-\ell}$. If $x^4$ lies on the line going through $x_0, x_1$, then $x_1$ is not attackable (since $f(x_0) = +1$ is not consistent with $f(x_1) = f(x_4) = -1$). Call the open half-space defined by the line going through $x_0, x_1$ that contains $x_4$, $H$. Without loss of generality, let $x_2$ be in $H$.

Then, apply Radon's theorem to $x_0, x_1, x_2, x_4$ to get the subsets $U', T'$ that have intersecting convex hulls. Without loss of generality assume $|U'| \leq |T'|$. Note that $U \neq \{x_0\}$, since $conv(\{x_1, x_2, x_4\})\setminus \{H\} = \{x_1\}$, and a similar argument shows that $U' \neq \{x_1\}$. Furthermore since all points in the convex hull of $x_0, x_1, x_2$ are labeled $+$, $U' \neq \{x_3\}$. Also, neither $U'$ nor $T'$ are $\{x_0, x_1\}$, since $conv(\{x_2, x_4\}) \subset H$, and similarly neither can be $\{x_0, x_4\}$, since for any $x \in conv(\{x_0, x_4\})$, either $x = x_0$ (and therefore is not in $conv(\{x_1, x_2\})$ (assuming $x_0$ is not attackable), or $w^\top x < w^\top x_0 \leq \min_{x\in conv(\{x_1,x_2\})} w^\top x' = \min_{x''\in\{x_1,x_2\}} w^\top x''$.

If the two sets are $\{x_0, x_2\}$ and $\{x_1, x_4\}$, then $x_1$ is not attackable, since setting it to $-$ is inconsistent with the other labels.

---

[3]In fact, it is not difficult to show that $\Phi_{-\ell}(\{a,b\}, S_T^{-\ell}) \geq 3$, but this is unnecessary for our purposes and does not asymptotically improve our bounds.

This leaves one remaining possibility, $U' = \{x_2\}$. In this case, notice that there exists a concept $f' \in \mathcal{F}$, with $f'(x_0) = -1$, $f'(x_2) = +1$, while compatible with any $S'^{-\ell}$ (that is, any subset of $\mathcal{X}$ that is labeled $-\ell$ by $f$), is incompatible with $f'(x_1) = f'(x_4) = -1$, so

$$\Phi_{-\ell}(\{x_0, x_2\}, S^{-\ell} \cup S' \cup \{x_1\}) < \Phi_{-\ell}(\{x_0, x_2\}, S^{-\ell} \cup S') \tag{3}$$

So, for every four samples from $S_{iid}^+$, either one of them is not attackable, or some triplet of them act as $x_0, x_1, x_2$ in eq. (3). Consider any set $A \subseteq S^\ell$ of non-attackable samples. By pigeon hole principle, for some $x \in A$, for $\frac{\binom{|A|}{4}}{|A|}$ of the elements of $A^4$ have $x$ act as $x_1$ in eq. (3). Notice that some pair $a, b \in S^\ell$ can vote for $x$ in at most $|A| - 3$ combinations from $A^4$ (once for every choice of $x_3$), so the number of unique pairs that act as $x_0, x_2$ while $x$ acts as $x_1$ in eq. (3) is at least

$$\frac{\binom{|A|}{4}}{|A|(|A| - 3)} = \frac{1}{12}\binom{|A| - 1}{2}$$

Then notice that, for any $S'$ that is labeled $-\ell$ by $f$,

$$\rho_{-\ell}(x, S^\ell, S^{-\ell} \cup S') = \sum_{a,b \in S}(\Phi_\ell(\{a,b\}, S^{-\ell} \cup S') - \Phi_\ell(\{a,b\}, S^{-\ell} \cup S' \cup \{x\})) \geq \frac{1}{12}\binom{|A| - 1}{2}$$

Since $x$ is not attackable (by the definition of $A$), this means that $\frac{1}{12}\binom{|A|-1}{2} < \alpha$, and so, $|A| < 48\sqrt{\alpha}$.

So, if we sum over both labels, there are at most $96\sqrt{\alpha}$ attackable samples.

Now notice that since non-adversarial samples are drawn iid, they are exchangeable, so any of the first $n$ such samples are equally likely to be the $n^{\text{th}}$, so the probability that the $n^{\text{th}}$ iid sample is attackable is at most $\frac{96\sqrt{\alpha}}{n}$. If the $n^{\text{th}}$ iid sample is *not* attackable, then the learner will not abstain on it. So, the expected number of abstentions on non-adversarial samples is at most

$$\sum_{n=1}^{T} \frac{2\alpha}{n} \leq 96\sqrt{\alpha}(\log T + 1) = O(\sqrt{\alpha}\log T)$$

$\square$

