# OpenReview forum: "Optimal Lower Bounds and New Upper Bounds for Sequential Prediction with Abstention"
_NeurIPS.cc/2025/Workshop/Reliable_ML — NeurIPS 2025 - Reliable ML Workshop_

### Official Review · Reviewer_SzJU · 2025-09-15
**Clear, solid theoretical paper on online learning**

**Rating:** 8
**Confidence:** 4

**Review:**

# Summary

The paper studies the problem of sequential prediction with abstentions, where the learner is asked to classify a stream of i.i.d. samples, with an adversary injecting out-of-distribution samples. The learner can choose to predict on a given sample or to abstain, which will not incur penalty. The total penalty is the sum of the number of misclassifications and abstentions on inliers. The previous work has given an algorithm that achieve $O(d^2 \log T)$ error for concept classes with VC dimension $d$, when the covariate distribution is known; and an algorithm that achieve $O(\sqrt{T})$ error for a specific class with VC dimension 1, when the covariate distribution is unknown.

The paper proves an $\Omega(\sqrt{T})$ lower bound on the error for a specific class with VC dimension 1 when the covariate distribution is unknown by a simple counting argument. Also, they present an algorithm for learning half-spaces in $\mathbb{R}^2$ when the covariate distribution is unknown that achieves $\tilde O(T^{2/3})$ error. The algorithm makes use of the past samples to vote for the current prediction, with a trade-off between the error from misclassifications and abstentions on inliers when the voting threshold varies.

# Strengths

The writing is well-structured and mostly clear to follow, and the proof sketches are intuitive. And the arguments are concise and simple.

For the $\Omega(\sqrt{T})$ lower bound, it shows separation between the settings with known and unknown distribution as the previous work gives a $O(d^2\log T)$ upper bound for any classes with VC dimension $d$.

For the upper bound, the paper presents the first algorithm for learning half-spaces in $\mathbb{R}^2$ in this setting, naturally generalizing the previous work and exploiting the geometric structure of the class.

# Weaknesses

The authors give a “hard” class that will incur $\Omega(\sqrt{T})$ error with unknown distribution, while the previous work gives an “easy” class that is learnable with $O(\sqrt{T})$ error with unknown distribution. It is not clear how the errors of learning other classes behave, when the covariate distribution is unknown. Also, while the authors give an algorithm for two dimensional half-spaces, they only conjecture the higher dimension cases.

# Suggestions

In the proof sketch for the upper bound, it would be good if the authors could explain how the geometric structure of the half-spaces is exploited.

In line 37, I do not see what “the existing algorithm” refers to. The previous work that achieves $O(\sqrt{T})$ error is for another class.

The definition the voting function will probably confusing first-time readers:
In line 119, the definition of $\mathcal{X}^*$ is not clear.
After line 123, in the definition of $\rho_\ell$, $(a, b) \in S$ should be $\\{a, b\\} \subseteq S$.

Some other typos:
In line 84, $X$ should be $\mathcal{X}$.
In line 128, the first letter of “each” should be capitalized.

---

### Official Review · Reviewer_5kmL · 2025-09-19
**A nice result on a relevant problem. Clear and well thought Lower Bound. Non-surprising upper bound extension.**

**Rating:** 7
**Confidence:** 4

**Review:**

1) Summary:
The authors address the issue of sequential prediction with abstention. In this problem, each input point is either drawn independently and identically (i.i.d.) from a fixed distribution over the domain, or selected by an adversary. If the point is drawn i.i.d., the learner incurs a penalty for misclassification or abstention. If the point is selected by an adversary, the learner incurs a misclassification penalty. The learner does not know whether a point was drawn independently or selected by an adversary. The objective is to minimize the total penalty.
The authors provide an Ω(√T) lower bound on the error for any learner facing a VC-dimension-1 class when the distribution is unknown. They also provide an O(T¹/²) algorithm for biased half spaces in R².
2) Strengths:
The problem is well-motivated, and the paper's results expand our understanding of the problem.
The paper's structure and phrasing are generally clear. The lower bound and the algorithm are presented well, and the proof sketches are helpful.
The topic of this paper is well within the scope of this workshop.
3) Weaknesses:
To my understanding, the upper bound for biased half spaces in R² is not based on a new understanding of the problem, but rather on the "exploitation of the geometric structure of half spaces," as the authors pointed out. Therefore, while it may be a useful step towards extending this framework to more general hypothesis classes, its theoretical significance is limited.
4) Suggestions for the authors:
The paper is generally well-written, but I had some trouble with the phrasing of the third bullet point regarding the lower bound and the sentence, "For our learner (and the...past examples with label l."

I also found some typos:
Line 107: class instead of class
On line 108, I think you mean b ∊ R.
Line 140: Forgot tilde.